# The Impact of Psilocybin on High Glucose/Lipid-Induced Changes in INS-1 Cell Viability and Dedifferentiation

**DOI:** 10.3390/genes15020183

**Published:** 2024-01-29

**Authors:** Esmaeel Ghasemi Gojani, Bo Wang, Dong-Ping Li, Olga Kovalchuk, Igor Kovalchuk

**Affiliations:** Department of Biological Sciences, University of Lethbridge, Lethbridge, AB T1K 3M4, Canada; esmaeel.ghasemigojan@uleth.ca (E.G.G.); bo.wang5@uleth.ca (B.W.); dongping.li@uleth.ca (D.-P.L.); olga.kovalchuk@uleth.ca (O.K.)

**Keywords:** psilocybin, diabetes, β-cell loss, β-cell dedifferentiation

## Abstract

Serotonin emerges as a pivotal factor influencing the growth and functionality of β-cells. Psilocybin, a natural compound derived from mushrooms of the *Psilocybe* genus, exerts agonistic effects on the serotonin 5-HT2A and 5-HT2B receptors, thereby mimicking serotonin’s behavior. This study investigates the potential impacts of psilocybin on β-cell viability, dedifferentiation, and function using an in vitro system. The INS-1 832/13 Rat Insulinoma cell line underwent psilocybin pretreatment, followed by exposure to high glucose-high lipid (HG-HL) conditions for specific time periods. After being harvested from treated cells, total transcript and cellular protein were utilized for further investigation. Our findings implied that psilocybin administration effectively mitigates HG-HL-stimulated β-cell loss, potentially mediated through the modulation of apoptotic biomarkers, which is possibly related to the mitigation of TXNIP, STAT-1, and STAT-3 phosphorylation. Furthermore, psilocybin exhibits the capacity to modulate the expression of key genes associated with β-cell dedifferentiation, including *Pou5f1* and *Nanog*, indicating its potential in attenuating β-cell dedifferentiation. This research lays the groundwork for further exploration into the therapeutic potential of psilocybin in Type II diabetes intervention.

## 1. Introduction

Diabetes mellitus is a persistent metabolic disorder wherein the body struggles to adequately manage its blood glucose levels. This results from insufficient insulin production or reduced responsiveness to insulin. Consequently, individuals with diabetes experience elevated blood glucose levels, disrupting the metabolism of carbohydrates, proteins, and fats. Additionally, diabetes is closely linked to various coexisting health conditions, including atherosclerosis, retinopathy, neuropathy, and nephropathy [1].

As of 2021, the global prevalence of diabetes, according to the International Diabetes Federation (IDF), was approximately 537 million people, projected to increase to around 643 million by 2030 and an estimated 783 million by 2045.

Obesity, sedentary lifestyle, and a westernized dietary are the most common risk factors of Type 2 diabetes mellitus (T2DM). Obesity, in particular, plays a prominent role in the progression of T2DM due to its association with insulin resistance development. Furthermore, obesity contributes to the dysfunction and reduction in pancreatic β-cells through undesirable outcomes of glucolipotoxicity [1].

In this regard, it has been observed that the increased levels of blood-free fatty acids (FFAs) and lipopolysaccharides (LPS) in obese people negatively impact β-cell functionality and survival through mechanisms such as endoplasmic reticulum (ER) stress initiation, the release of proinflammatory cytokines, upregulation of inducible nitric oxide synthase (iNOS), and impaired insulin signaling [2]. Sustained endoplasmic reticulum (ER) stress in β-cells, induced by heightened insulin need in the phase of insulin resistance, exacerbates their dysfunction and deterioration. Recent studies have suggested that individuals with T2DM experience a significant decline in β-cells [3].

Pancreatic β-cells undergo differentiation from embryonic stem cells, involving the activation or repression of unique genetic markers and effectors, to acquire distinct features and functions. However, mature β-cells can undergo dedifferentiation, losing their specialized characteristics and reverting to a less differentiated state. This dedifferentiation is believed to contribute to the reduction in an effective β-cell population in diabetes. It involves the downregulation of genes typically expressed in β-cells and the upregulation of genes normally suppressed in healthy β-cells. These changes lead to an altered β-cell phenotype and impaired insulin secretion [4,5,6,7]. Glucolipotoxicity, resulting from high glucose and high lipid (HG-HL) levels, exerts a crucial influence on β-cell dedifferentiation. The precise mechanisms involved, including oxidative stress, endoplasmic reticulum stress, inflammation, and hypoxia, are actively investigated. The role of specific transcription factors like POU domain, class 5, transcription factor 1 (Pou5f1), Nanog homeobox (Nanog), pancreatic and duodenal homeobox 1 (PDX1), v-maf musculoaponeurotic fibrosarcoma oncogene homolog A (MafA), neurogenic differentiation 1 (NEUROD1), NK6 homeobox 1 (Nkx6.1), and forkhead box protein O1 (FOXO1) in β-cell dedifferentiation has been well studied [4,8].

Serotonin, also referred to as 5-hydroxytryptamine (5-HT), is acknowledged as a crucial modulator of both β-cell growth and insulin release. The coexistence of serotonin with insulin in the same vesicles within pancreatic β-cells has been widely recognized, affirming its significance in the regulation of glycemia [9,10]. In the islets of the pancreas, serotonin functions through the autocrine/paracrine signaling pathway by binding to the 5-HT receptors, stimulating the secretion of insulin in the presence of glucose (glucose-stimulated insulin secretion or GSIS) [10]. Additionally, during the middle stage of gestation, there is an upregulation of the 5-HT2B receptor expression. This increase in receptor expression promotes β-cells growth, resulting in an expansion of the β-cells population [11]. This observation highlights the significant involvement of this receptor and its downstream pathway(s) in the modulation of β-cell growth [12,13].

Psychedelic compounds interact with serotonin receptors and their different subtypes located throughout the brain. These receptors play a role in regulating various processes such as emotions, moods (including anxiety and aggression), cognition, sexual behavior, learning, memory, and appetite [14,15]. It is worth noting that these receptors are not limited to the central nervous system but also exist in the peripheral nervous system [16].

Psychedelic compounds, like psilocybin discovered in the *Psilocybe* genus of mushrooms, interact with serotonin receptors in the brain, influencing emotions, moods, cognition, sexual behavior, learning, memory, and appetite. Psilocybin, known for its favorable safety profile, acts as an agonist, particularly on the 5-HT2A receptor. Its ability to modulate 5-HT2A receptor activation and downstream gene processes categorizes it as a safe serotonin receptor agonist [17,18]. 

In this study, our objective was to reveal how psilocybin influences the loss and dedifferentiation of β-cells induced by HG-HL. Our findings indicate that psilocybin effectively opposes the loss of β-cells triggered by HG-HL by reducing apoptosis. Moreover, our research offers indications that psilocybin might impede the dedifferentiation of β-cells induced by HG-HL. 

## 2. Materials and Methods

### 2.1. Cellular Culturing and Experimental Treatments

The INS-1 832/13 Rat Insulinoma Cell line (Catalog Number: SCC207, sourced from EMD Millipore Corporation, Temecula, CA, USA) is a variant of INS-1, selected for its strong glucose-stimulated insulin secretion (GSIS) capabilities [19]. It features a human insulin expression cassette, facilitating the production and release of both rat and human insulin. The presence of a human insulin expression cassette ensures the consistent secretion of human insulin over extended passages under selective conditions [20]. The INS-1 832/13 cellular model also serves as a valuable system for exploring the intricacies of cellular insulin release, storage, and production mechanisms (EMD Millipore). The cells were cultivated in a RPMI 1640 medium (Catalog Number: 350-060-CL, procured from Wisent Inc., Saint-Jean-Baptiste, QC, Canada). The medium was enriched with 10% heat-treated high-quality fetal bovine serum (FBS) (Catalog Number: 10082147, procured from Fisher Scientific Company, Ottawa, ON, Canada) and specific concentrations of L-Glutamine (Catalog Number: TMS-002-C), sodium pyruvate (Catalog Number: S8636), HEPES (Catalog Number: TMS-003-C), β-mercaptoethanol (Catalog Number: ES-007) (all acquired from EMD Millipore Corporation, Temecula, CA, USA), and glucose, followed by incubating in the appropriate conditions. Cell passages 5–10 were utilized for all experiments, with a routine medium change every 48 h.

Subsequently, cells underwent treatment with a specified dosage of psilocybin (CAS No. 520-52-50, received from Applied Pharmaceutical Innovation, Edmonton, AB, Canada) for 2 h. In the present study, all experiments were conducted using a 10 µM concentration of psilocybin, a dosage determined based on the outcomes from our previous results [21]. Pretreatment with psilocybin was followed by exposure to a combination of high glucose (HG) conditions (25 mM glucose) and high lipid (HL) conditions (400 μM/250 μM palmitic acid) for the specific time periods. Based on our optimization experiments, we employed two distinct experimental conditions for the induction of apoptosis (400 μM PA + 25 mM glucose for 48 h), investigated through Western blot analysis, and dedifferentiation (250 μM PA + 25 mM glucose for 25 h), tested by quantitative real-time polymerase chain reaction (qRT-PCR).

A glucose stock solution with a concentration of 1.0 M was prepared by dissolving glucose in a medium, followed by filter sterilization, and then stored at 4 °C. For the preparation of the palmitic acid (PA) stock, PA was dissolved in 100% ethanol with heating at 70 °C. Subsequently, the dissolved PA was mixed with sterile 10% BSA through two rounds of heating at 55 °C for 15 min and mixing. The PA stock was aliquoted and stocked at −20 °C. Before its use, the PA stock was reheated at 55 °C for 15 min. Psilocybin stock was freshly prepared using a medium.

### 2.2. Cell Viability Assessment

To evaluate how psilocybin influences the viability of β-cells in conditions of HG-HL, we utilized an MTT assay. Initially, cells were cultured at a density of 5 × 10^5^ cells per well in 100 μL of cultivation medium, which contained psilocybin, for a 2 h pre-treatment. Subsequently, cells were cultured in a medium with psilocybin and/or HG (25 mM Glucose)-HL (400 μM PA) conditions for the following 48 h. Following the cultivation period, 10 μL of an MTT labeling reagent (3-(4,5-Dimethylthiazol-2-yl)-2,5-diphenyltetrazolium bromide) (Catalog Number: 11465007001, MilliporeSigma Canada Ltd., Oakville, ON, Canada) was introduced into each well. The microplate was placed in an incubator for the next 4 h. Subsequently, 100 μL of a solubilization solution, composed of SDS 10% and hydrochloric acid (HCL), was introduced to each well, and the plate was kept in an incubator overnight. The optical density at a wavelength of 595 nm was assessed utilizing a plate reader (FLUOstar Omega, BMG LABTECH, Offenburg, Germany).

### 2.3. Protein Immunoblotting

To initiate protein separation, the protein samples were blended with an equimolar amount of 1X loading buffer and then exposed to heating at 95 °C for 5 min. Following the separation of 50–100 μg of total protein with SDS-PAGE, the resolved proteins were electro-transferred onto polyvinylidene difluoride (PVDF) membranes, specifically employing Amersham Hybond^®^ P membranes (RPN2020F) obtained from GE Healthcare (Oakville, ON, Canada). Preceding an extended overnight incubation with primary antibodies at 4 °C, the membranes underwent blocking, employing PBST (phosphate-buffered saline with Tween) supplemented with 5% milk. Subsequent to washing the membranes three times with PBST, they were subjected to secondary antibodies for a period of two hours at room temperature. This was followed by another series of PBST washes. Immunoreactivity was identified using peroxidase-conjugated antibodies, and the ensuing reaction was visualized through the ECL Plus Western blotting detection system (GE Healthcare, Oakville, ON, Canada). For imaging the Western blot bands, an Amersham Imager 600 RGB was utilized. Band intensities were measured through ImageJ analysis and subsequently normalized against the intensity of housekeeping proteins. Comprehensive details of the primary antibodies are outlined in Appendix A.

### 2.4. Gene Expression Profiling

The process of RNA extraction from cellular material began with the use of TRIzol^®^ Reagent (15596018, Invitrogen, Life Technologies Inc., Burlington, ON, Canada), following the producer’s recommended procedures. The RNA quantity was then determined using a Nanodrop 2000c (ThermoFisher Scientific, Waltham, MA, USA). Subsequently, 1 μg of total RNA was utilized for the synthesis of complementary DNA (cDNA) with the iScript™ Reverse Transcription Supermix (1708841, BioRad Laboratories, Saint-Laurent, QC, Canada), following the manufacturer’s instructions. The resulting cDNA product served as a template for qRT-PCR, with each reaction containing 1 μL of cDNA. The qRT-PCR reactions were carried out using the SsAdvancedTM Universal Inhibitor-Tolerant SYBR Green Supermix (1725017, Bio-Rad Laboratories, Saint-Laurent, QC, Canada). The primers for the qPCR experiments were obtained from Eurofins (Ottawa, ON, Canada) (Appendix A).

### 2.5. Glucose Stimulated Insulin Secretion (GSIS) Assay

To assess the influence of psilocybin on β-cell responsiveness under HG-HL, a GSIS assay was executed. The β-cells were cultured at a density of 0.5 × 10^6^ cells per well in a 24-well plate and allowed to grow for a duration of 2 days. Afterwards, the cells underwent pre-treatment with a media solution containing psilocybin for 2 h before being exposed to HG-HL conditions (400 μM palmitic acid + 25 mM glucose) for a period of 48 h. The GSIS assay utilized HEPES Balanced Salt Solution (HBSS) with specific salt concentrations and a BSA at a pH of 7.2. The cells underwent a dual wash with HBSS containing 2.5 mM glucose, with the second wash extending for 1 h. Each treatment condition was assigned to two wells. In one well, the treatment including HBSS comprising 10 μM psilocybin and 2.5 mM glucose (depicting normal glucose), while the other well incubated with HBSS containing 10 μM psilocybin and 16.5 mM glucose (depicting high glucose levels). Each well experienced a 2 h incubation period, after which the collected solution underwent analysis using insulin ELISA.

The ELISA assay was carried out using the Rat/Mouse Insulin ELISA kit (EZRMI-13K, obtained from Sigma Aldrich, Oakville, ON, Canada), adhering to the producer’s guidelines. Plate absorbance readings were assessed at 450 nm and 590 nm with a plate reader (SpectraMax i3x Multi-Mode Microplate Reader, Molecular Devices, San Jose, CA, USA). The subtraction of absorbances at 590 nm from those at 450 nm was performed. To construct a standard curve, the kit-provided standards were utilized, and the resulting equation from the standard curve was applied to convert the absorbance values into insulin amounts.

### 2.6. Caspase 3 and Caspase 7 Activity

The potential impact of psilocybin on the reduction in Caspase 3 and Caspase 7 activity in the β-cells exposed to HG-HL conditions was evaluated using the Caspase-Glo^®^ 3/7 3D Assay kit (Cat: G8981) obtained from Promega Corporation (Madison, WI, USA). β-cells were plated at a concentration of 5 × 10^4^ cells per well in a 96-well plate. After 48 h, the samples were exposed to psilocybin for 2 h. Afterwards, the cells were subjected to a medium containing psilocybin and/or HG-HL for the subsequent 48 h. Upon completion of the incubation phase, the plate was taken out and allowed to reach room temperature before adding Caspase-Glo^®^ 3/7 3D Reagent to each well. The contents were thoroughly mixed and incubated for a minimum of 30 min before measuring luminescence using a plate-reading luminometer (SpectraMax i3x Multi-Mode Microplate Reader, Molecular Devices, San Jose, CA, USA).

### 2.7. Statistics

The gathered data underwent rigorous statistical scrutiny, employing the one-way analysis of variance (ANOVA) as the initial step. Post hoc analyses were then conducted using both the Tukey and Dunnett’s tests to make meaningful comparisons between mean values across different experimental groups. The statistical analyses were executed utilizing GraphPad Prism 8.0 software, which is renowned for its robust statistical tools and capabilities. Significance was set at a threshold of *p* < 0.05, ensuring that only the results with a probability of occurrence lower than 5% were considered statistically significant.

## 3. Results

### 3.1. 10 µM Psilocybin Mitigates HG-HL-Induced β-Cell Reduced Viability

To examine the effects of psilocybin on β-cell loss caused by HG-HL conditions (400 μM PA + 25 mM glucose), an MTT assay was conducted, involving four treatment groups: Control (Ct), Control + Psilocybin 10 μM (Ct + PSI 10), HG-HL, and HG-HL + Psilocybin 10 μM (HG-HL + PSI10). The administration of 10 μM psilocybin to β-cells resulted in a significant improvement in cell survival under HG-HL conditions (Figure 1). Importantly, the initial cell viability was consistent across all wells of the 96-well plate, showing equal cell loading on day 0.

### 3.2. Psilocybin Decreases the Levels of Apoptotic Biomarkers in HG-HL-Challenged β-Cells

Through Western blot analysis, we observed that 10 μM psilocybin effectively lowered the increased levels of Pro-Caspase-9, Cleaved-Caspase-9 (C-Caspase-9), Pro-Caspase-7, C-Caspase-7, C-Caspse-3, C-PARP, Bcl-2, Bim and BAX proteins in β-cells subjected to HG-HL conditions (400 μM PA + 25 mM glucose for 48 h) (Figure 2a). This reduction suggests the potential of psilocybin to mitigate apoptosis and thereby prevent HG-HL-induced β-cell loss. The results of the Caspase3/7 activity kit further supported the Western blot findings of the apoptotic proteins in response to psilocybin (Figure 2c).

### 3.3. Psilocybin Mitigates the Phosphorylation of STAT-1 and STAT-3 

Signal transducer, activator of transcription 1 (STAT-1), and STAT-3 are transcription factors that play crucial roles in the cellular responses to various signals, including those that regulate apoptosis [22,23]. According to our results, administration of psilocybin mitigates the elevated levels of phospho-STAT-1 (P-STAT-1), P-STAT-3, and total-STAT-3 in HG-HL-stimulated β-cells (Figure 3). 

### 3.4. Psilocybin Did Not Improve the Impaired GSIS Caused by HG-HL Conditions

The process of GSIS regulates the blood glucose levels and is controlled by pancreatic β-cells. Hyperglycemia and hyperlipidemia conditions impair GSIS, contributing to the development and progression of T2DM and metabolic syndrome. In this study, our objective was to assess the potential of psilocybin in alleviating HG-HL-induced GSIS impairment. We observed that under HG-HL conditions, psilocybin does not notably enhance the compromised GSIS (Figure 4).

### 3.5. The Response of PDX-1, FOXO1, Phospho-FOXO1 (P-FOXO1), and TXNIP Proteins to 10 μM Psilocybin in HG-HL-Induced Β-Cells

PDX-1 is a crucial transcription factor responsible for the formation and operation of pancreatic β-cells [24]. Our findings reveal that psilocybin effectively mitigated the reduced levels of this transcription factor induced by HG-HL conditions (400 μM palmitic acid + 25 mM glucose for 48 h) in β-cells (Figure 5).

FOXO1, another transcription factor abundantly present in various tissues, including pancreatic β-cells, contributes significantly to the functionality and viability of β-cells. The activity and subcellular localization of FOXO1 are regulated by a critical post-translational modification, namely phosphorylation. Phosphorylation at multiple sites leads to the sequestration of FOXO1 in the cytoplasm, suppressing its transcriptional activity [25]. Our results indicate that the psilocybin treatment led to a reduction in both phosphorylated and unphosphorylated forms of FOXO1 in HG-HL-induced β-cells (Figure 5).

Thioredoxin interacting protein (TXNIP) is a key player in the apoptosis process. As a suppressive controller of thioredoxin, elevated expression of this factor has been linked to triggering apoptosis in diverse cell types [26]. The Western blot analysis of TXNIP in β-cells challenged by HG-HL conditions, following exposure to psilocybin, displayed a significant decrease in the induced levels of this protein (Figure 5). 

### 3.6. Psilocybin Administration Downregulated Nanog and Pou5f1 in the HG-HL-Challenged β-Cells 

To investigate the impact of 10 μM psilocybin on β-cell dedifferentiation caused by HG-HL conditions (250 μM PA + 25 mM glucose for 24 h), we examined the expression of key biomarkers. The qRT-PCR analysis included four experimental treatments: Ct, Ct + PSI10, HG-HL, and HG-HL + PSI10. Under the HG-HL conditions, there was a reduction in the transcript levels of *PDX-1*, *Ins1*, *Ins2*, *NEUROD1*, *Slc2A2*, *FOXO1*, and *MafA*. Psilocybin did not restore the decreased transcript levels of these genes. HG-HL exposure significantly increased the expression of *Nanog* and *Pou5f1* genes, while the addition of 10 μM psilocybin downregulated the mRNA levels of *Nanog* and *Pou5f1* (Figure 6).

## 4. Discussion

This study focused on examining the potential protective effects of psilocybin against HG-HL-induced loss and dedifferentiation of β-cells. Through our MTT assay, it was observed that treatment with 10 μM psilocybin was able to restore the viability of HG-HL-challenged β-cells (Figure 1). Western blot analysis of the key apoptotic biomarkers showed that psilocybin may exert this protective effect by mitigating apoptosis in HG-HL-challenged β-cells (Figure 2a). Additionally, the results from the Caspase-3/Caspase-7 activity kit provided further evidence of this protective impact (Figure 2b). 

Apoptosis occurs through two primary signaling routes: the extrinsic pathway and the intrinsic pathway. The extrinsic pathway is initiated by external signals and led to the stimulation of death receptors on the cell surface, whereas the intrinsic pathway is activated in response to internal stress signals, such as DNA damage or issues with mitochondrial function, which may be caused by high glucose and high lipid conditions. Both pathways ultimately lead to the activation of executioner caspases, including Caspase-3 and Caspase-7 [27]. Within the intrinsic pathway, a cascade of intracellular stress signals induces the permeability of the mitochondrial outer membrane, facilitating the release of pro-apoptotic proteins, like cytochrome c. This, in turn, activates caspases, propelling the apoptotic process [28]. Biomarkers associated with intrinsic apoptosis encompass proteins from the Bcl-2 family, such as Bax, Bak, Bim, and Puma, which regulate mitochondrial outer membrane permeabilization. Additional intrinsic apoptosis biomarkers include the release of cytochrome c from the mitochondria and the activation of caspase-9 [29,30]. 

HG-HL have been linked to the induction of endoplasmic reticulum (ER) stress and the activation of intrinsic apoptosis in β-cells [31]. Our experimental findings, as depicted in Figure 2, reveal that treatment with psilocybin significantly mitigates intrinsic apoptotic biomarkers, including Bax, Bim, Pro-Caspase-9, and C-Caspase-9. Notably, under HG-HL conditions, the level of the anti-apoptotic biomarker Bcl-2 is elevated in β-cells, but this elevation is reversed upon treatment with 10 μM psilocybin (Figure 2). It is essential to note that despite the acknowledged anti-apoptotic role of Bcl-2, evidence suggests that Bcl proteins may regulate the pancreatic β-cell response to glucose, implying their role as integrators of cell death and the physiology in these cells. Indeed, studies have demonstrated that the chemical and genetic loss of function of antiapoptotic Bcl-2 and Bcl-xL significantly enhance glucose-dependent metabolic, Ca^2+^ signals, and subsequently, GSIS response in primary pancreatic β-cells [32]. This may provide support for our results, explaining why HG-HL conditions lead to a reduction in the levels of Bcl-2 in untreated β-cells.

The interplay between STAT-1 and STAT-3 in β-cell apoptosis is intricate. Typically, STAT-1 promotes apoptosis by upregulating the expression of cell surface death receptor family members, their ligands, caspases, and inducible nitric oxide synthase (iNOS) [22,33,34]. Our findings indicate that psilocybin attenuates the heightened level of P-STAT-1 in HG-HL-induced β-cells, suggesting a potential target for psilocybin to mitigate apoptosis in β-cells challenged by HG-HL (Figure 3). 

Regarding the role of STAT-3 in β-cell apoptosis, conflicting reports exist. While some studies suggest that STAT-3 is generally recognized for its antiapoptotic action and its involvement in promoting cell proliferation in β-cells [35,36], others propose a relationship between activated STAT3 and the initiation of apoptotic responses in β-cells [37,38].

Under the HG-HL conditions employed in this study (48 h incubation with 400 μM PA + 25 mM glucose), P-STAT-3 and total STAT3 (T-STAT-3) levels increased, with mitigation observed after treatment with 10 μM psilocybin (Figure 3). These results suggest that the phosphorylation of STAT-3, in conjunction with STAT-1 phosphorylation, may serve as potential targets for psilocybin to exert its mitigative impact on the downregulation of apoptotic proteins.

TXNIP is a pivotal regulatory protein involved in diverse cellular processes, including the modulation of oxidative stress, inflammation, and apoptosis. Its crucial role in regulating the survival and function of β-cells is well established. Elevated levels of TXNIP have been demonstrated to trigger apoptosis in β-cells, while a deficiency in TXNIP has shown protective effects against both type 1 and type 2 diabetes by enhancing β-cell survival [39,40]. Our findings reveal that treatment with 10 μM psilocybin significantly reduced the induced levels of TXNIP in HG-HL-induced β-cells. This highlights an additional target for psilocybin, suggesting its potential to mitigate apoptosis in β-cells stimulated by HG-HL (Figure 3).

It should be mentioned that our findings, as detailed in the Appendix A regarding the serotonin receptor 2B (SR-2B), indicate that the administration of 10 μM psilocybin reduces the levels of this receptor in HG-HL-induced INS-1 cells (Appendix A). This discovery suggests a potential upstream target for psilocybin in influencing the modulation of apoptosis-related biomarkers.

In this investigation, we also examined the response of specific transcripts/proteins implicated in the dedifferentiation of β-cells. PDX-1 and FOXO1 stand out as critical transcription factors integral to the regulation of β-cell function. The intricate interplay between PDX-1 and FOXO1 plays a pivotal role in preserving β-cell function and ensuring glucose homeostasis. Disruptions in the regulation of these transcription factors are linked to β-cell dysfunction and the onset of diabetes. The delicate balance between the activities of PDX-1 and FOXO1 is precisely calibrated to guarantee appropriate insulin secretion and β-cell survival, adapting to the dynamic metabolic demands [41]. In our study, psilocybin further decreased the levels of PDX-1 and FOXO1 proteins in β-cells exposed to the HG-HL conditions, optimized for the induction of apoptosis (Figure 5). This suggests that the influence of psilocybin on β-cell loss and dedifferentiation under identical experimental conditions may vary. Specifically, it mitigates β-cell apoptosis induced by HG-HL conditions, concurrently reducing β-cell-specific biomarkers, and thereby increasing β-cell dedifferentiation.

FOXO1 phosphorylation, especially at Ser256, reduces its positive charge within the DNA binding domain. This decreases the transcriptional activity of FOXO1 [25,42]. According to our findings, the administration of psilocybin led to a decrease in the phosphorylation of FOXO1 in HG-HL-induced β-cells, consequently enhancing its transcriptional activity (as illustrated in Figure 5), providing initial evidence regarding the impact of psilocybin on the post-translational modification of FOXO1.

We also examined the impact of 10 μM psilocybin on the expression of key biomarkers specific to β-cells and also progenitor cells in β-cells exposed to optimized HG-HL conditions. The administration of psilocybin did not show any significant effects on the level of *PDX-1*, *FOXO1*, *Slc2A2* and *MafA* transcripts and even reduced the levels of *NEUROD1*, *Ins-1*, and *Ins-2* transcripts in HG-HL-induced β-cells (Figure 6). The observed outcomes, in conjunction with the Western blot results for PDX-1 and FOXO1, indicate that psilocybin may not exert a positive impact on the modified identity of β-cells induced by HG-HL through the restoration of β-cell-specific biomarker levels.

The transcription factors Pou5f1, Nanog, and L-MYC, which are associated with progenitor cell development, exhibit increased expression levels in individuals with T2DM [43]. In contrast to β-cell-specific biomarkers, the qRT-PCR results for *Pou5f1* and *Nanog* indicate that the administration of psilocybin can reduce the elevated expression levels of these transcription factors in β-cells induced by HG-HL conditions (Figure 6). Therefore, unlike the observed effects of psilocybin on β-cell-specific biomarkers, these findings may provide initial evidence suggesting the potential of psilocybin in mitigating β-cell dedifferentiation.

Our results demonstrate that the administration of psilocybin did not effectively restore the impaired GSIS (Figure 4), observed under HG-HL conditions. This lack of restoration may be attributed to the specific response of transcripts involved in glucose uptake, such as *Slc2A2*, and the transcription factor *MafA*, responsible for driving the transcription of *Slc2A2,* to psilocybin treatment (Figure 6). Additionally, none of the insulin gene isoforms exhibited a positive response to psilocybin administration in HG-HL-challenged β-cells. This lack of response may be associated with the reaction of *NEUROD1*, *PDX-1*, and *MafA* genes, collectively involved in the transcriptional process of insulin genes, to psilocybin treatment (Figure 6). 

## 5. Conclusions

In this study, we have presented compelling evidence showcasing the capacity of psilocybin to mitigate the loss of β-cells induced by HG-HL conditions. The protective effect of psilocybin against β-cell loss is attributed to its ability to alleviate intrinsic apoptosis. This is mediated through the reduction in TXNIP and the attenuation of STAT-1 and STAT-3 activation, as illustrated in our results. Furthermore, we found that psilocybin influences HG-HL-induced apoptosis and dedifferentiation in different ways. While it mitigates HG-HL-induced apoptosis in β-cells, it does not affect or may even exacerbate dedifferentiation in β-cells induced by HG-HL, as shown by the response of β-cell-specific biomarkers to psilocybin in HG-HL-induced β-cells. However, the potential inhibitory effect on the response of Nanog and Pou5f1 in HG-HL-challenged β-cells may open avenues for further exploration into the anti-β-cell dedifferentiation impact of psilocybin.

## 6. Limitations and Future Directions

The findings from this study provide preliminary evidence highlighting the potential therapeutic effects of psilocybin against the decrease in β-cell viability and dedifferentiation, forming a basis for further exploration into its potential antidiabetic effect. Future investigations are recommended to validate these findings using isolated pancreas models and animal models of T2DM. Additionally, exploring the impact of psilocybin on other forms of programmed cell death, such as pyroptosis, would be important.

In this study, we examined the impact of psilocybin on the mRNA levels of various transcription factors and proteins involved in β-cell dedifferentiation. It is recommended to extend the investigation to assess the response of the corresponding proteins and the expression of relevant genes regulated by these transcription factors to further validate our qRT-PCR results.

In our current study, we identified STAT1, STAT3, and TXNIP as potential downstream targets for psilocybin, contributing to the mitigation of apoptotic responses. The IL-6/JAKs axis typically facilitates the phosphorylation and subsequent activation of STATs. To determine whether this axis is implicated in the mitigative effects of psilocybin on apoptosis, it is recommended to initially study the response of this axis to psilocybin in HG-HL-induced β-cells.

Given psilocybin’s potential influence as a serotonin receptor agonist on cell proliferation, further research is proposed to investigate the response of the cell cycle and relevant biomarkers to psilocybin administration. As psilocybin has agonistic effects on the 5HT2A and 5HT2B receptors, it is advisable to examine the response of these receptors and other components within the serotonin signaling pathway to identify potential downstream targets.

It is also recommended to confirm the involvement of STAT3 and SR-2B in the modulatory effects of psilocybin on apoptosis responses by inhibiting their activation and induction.

## Figures and Tables

**Figure 1 genes-15-00183-f001:**
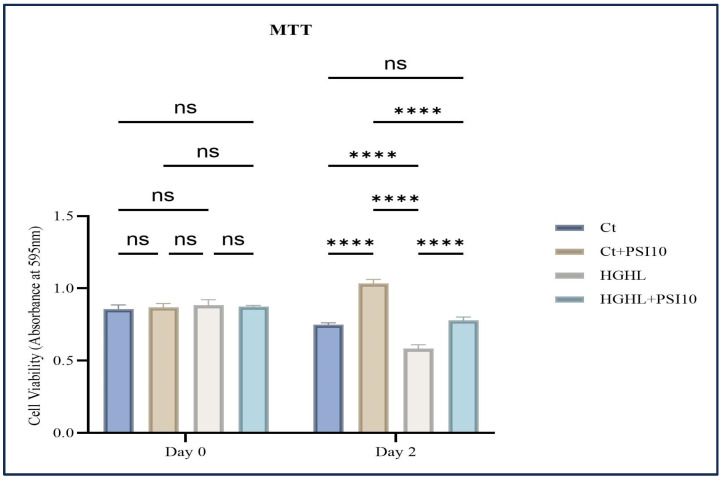
The impact of 48 h treatment with 10 μM psilocybin on the survival of β-cells challenged by HG-HL. INS-1 cells were pretreated with 10 μM psilocybin, followed by induction with HG-HL (400 μM PA + 25 mM glucose) for the next 48 h. Afterward, the cells were incubated with the MTT reagent for 4 h, followed by the incubation with MTT solubilizing buffer overnight. The results are depicted as the mean value with standard deviation, based on three experiments (N = 3). Abbreviations used are Ct (control), PSI (psilocybin), and HG-HL (high glucose + high lipid). Significance is denoted by asterisks, with four indicating *p* < 0.0001, while “ns” denotes non-significance.

**Figure 2 genes-15-00183-f002:**
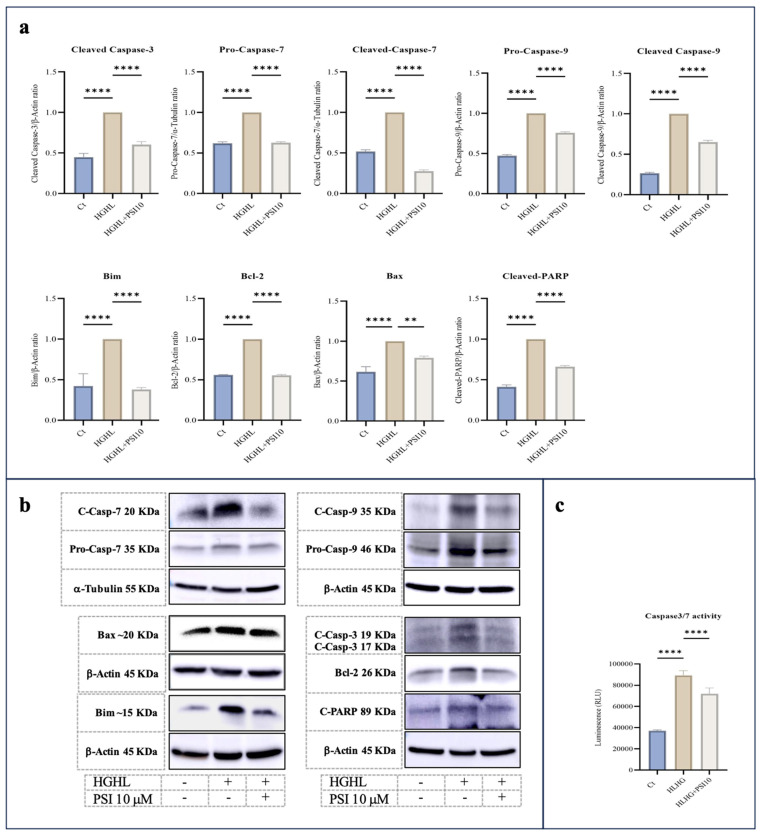
The immunoblot analysis of apoptotic biomarkers in HG-HL-challenged β-cells in response to psilocybin. (**a**) INS-1 cells were exposed to 10 μM psilocybin, followed by incubation with HG-HL for the next 48 h. Subsequently, the treated cells were used for protein extraction for Western blot analysis. The response of Pro-Caspase-9, C-Caspase-9, Pro-Caspase-7, C-Caspase-7, C-Capse-3, C-PARP, Bcl-2, Bim and Bax to 10 μM psilocybin in HG-HL-challenged β-cells, with normalization to housekeeping proteins. (**b**) Depicted are Western blot images of the apoptotic biomarkers. (**c**) Caspase 3/7 activity assay performed on β-cells challenged by HG-HL and treated with 10 μM psilocybin showed the mitigative impact of psilocybin on the elevated Caspase-3 and Caspase-7 activity in HG-HL-challenged β-cells. The results are depicted as the mean value with standard deviation, based on three measurements (N = 3). Abbreviations used are Ct (control), PSI (psilocybin), and HG-HL (high glucose + high lipid). Significance is denoted by asterisks, with two implying *p* < 0.01 and four indicating *p* < 0.0001.

**Figure 3 genes-15-00183-f003:**
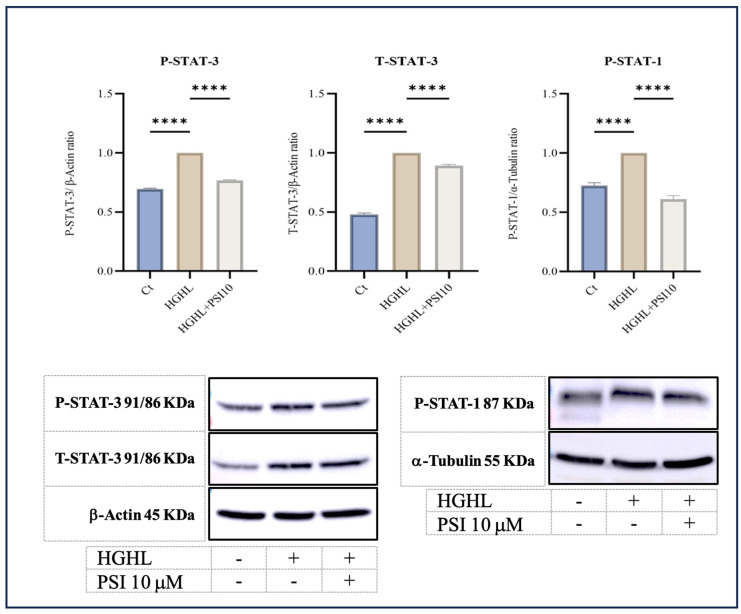
The immunoblot analysis of P-STAT-3, T-STAT-3, and P-STAT-1 in HG-HL- β-cells in response to 10 μM psilocybin. The administration of 10 μM psilocybin resulted in significant reduction in the levels of P-STAT-3, T-STAT-3, and P-STAT-1 in HG-HL-challenged β-cells. Following band quantification using ImageJ, the obtained results were normalized to housekeeping proteins, including α-Tubulin and β-Actin. The results are depicted as the mean value with standard deviation, based on three measurements (N = 3). Abbreviation: Ct (control), PSI (psilocybin) and HG-HL (high glucose + high lipid). Significance is denoted by asterisks, with four implying *p* < 0.0001.

**Figure 4 genes-15-00183-f004:**
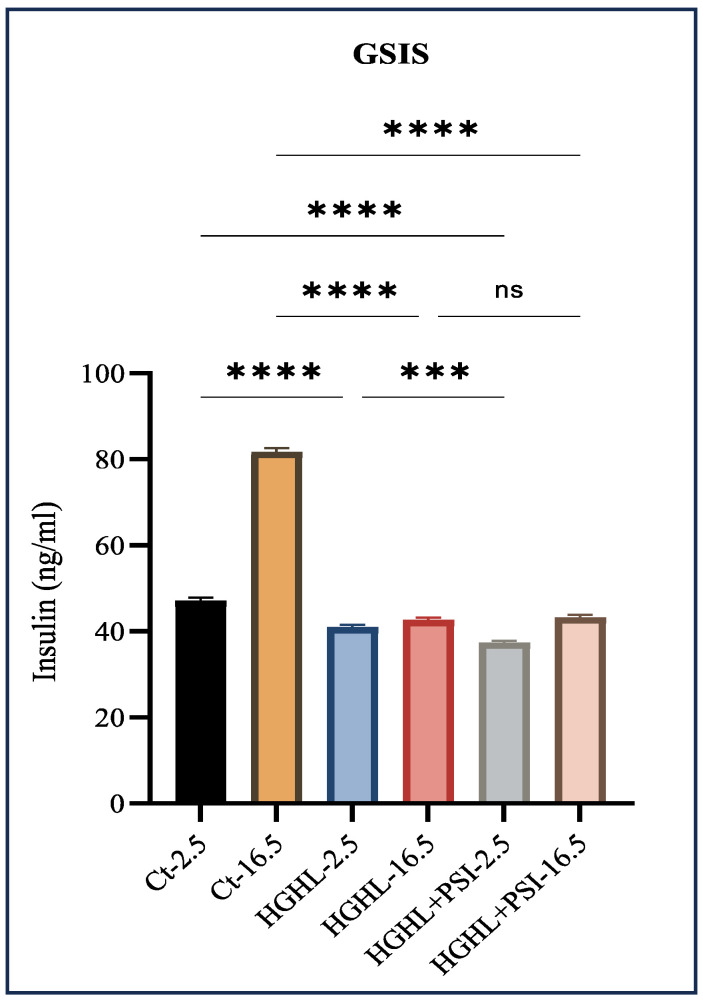
The impact of 10 μM psilocybin on GSIS of HG-HL-challenged β-cells. GSIS response of HG-HL-induced β-cell to psilocybin in media comprising 2.5 mM glucose and 16.5 mM glucose. Psilocybin did not exhibit any positive effects on the impaired GSIS in HG-HL-stimulated β-cells. The results are depicted as the mean value with standard deviation, based on three measurements (N = 3). Abbreviations used are Ct (control), PSI (psilocybin), and HG-HL (high glucose + high lipid). Significance is denoted by asterisks, with three implying *p* < 0.001 and four indicating *p* < 0.0001, while “ns” denotes non-significance.

**Figure 5 genes-15-00183-f005:**
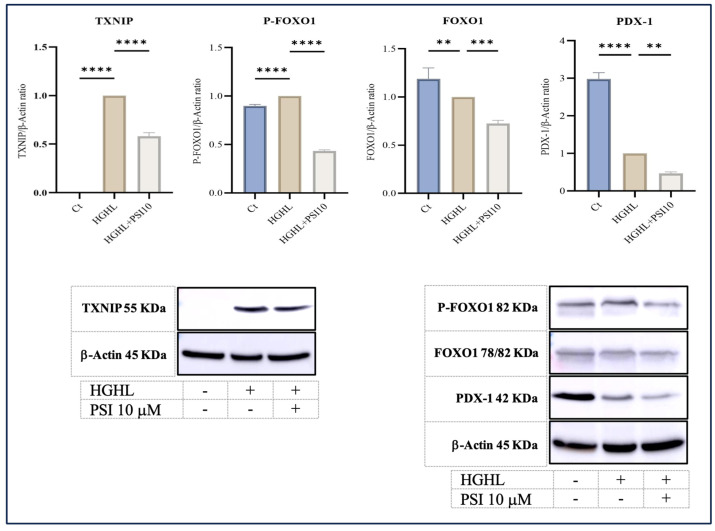
The alterations of P-FOXO1, FOXO1, TXNIP, and PDX-1 levels in HG-HL-stimulated β-cell in response to psilocybin. The administration of 10 μM psilocybin resulted in the reduction in P-FOXO1, FOXO1, TXNIP, and PDX-1 levels in HG-HL-induced β-cells. The findings are illustrated as the average value with standard deviation, derived from three measurements (N = 3). Abbreviations include Ct (control), PSI (psilocybin), and HG-HL (high glucose + high lipid). Significance is indicated by asterisks, where two asterisks indicate *p* < 0.01, three indicate *p* < 0.001, and four indicate *p* < 0.0001.

**Figure 6 genes-15-00183-f006:**
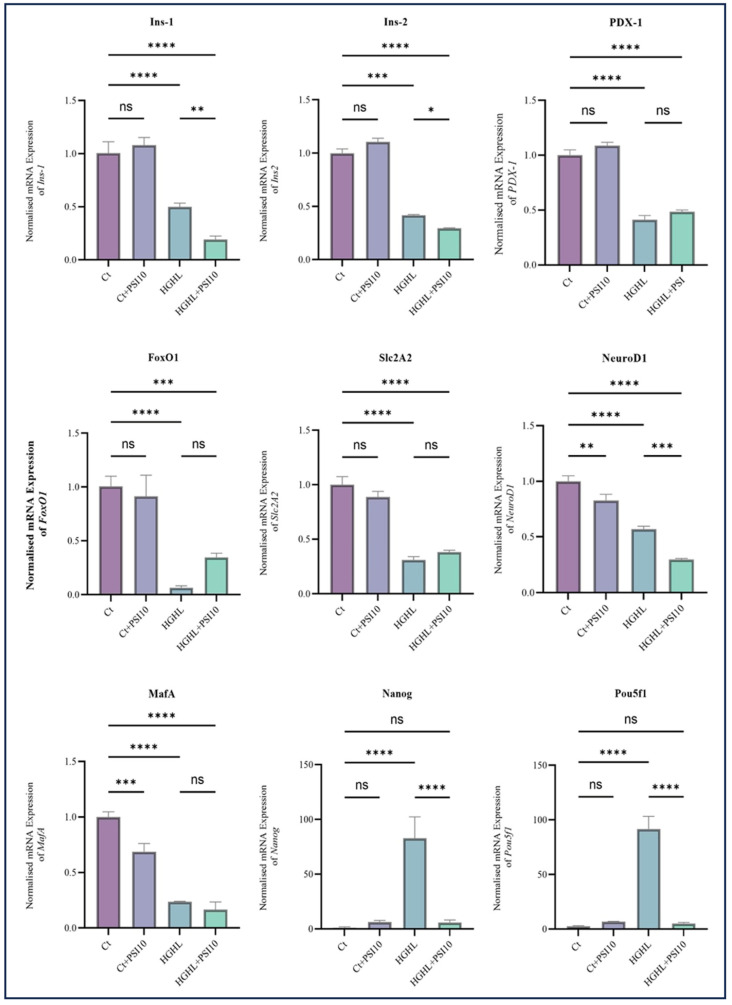
The qPCR analysis assessed the expression of key dedifferentiation-associated transcripts in both HG-HL-stimulated and unstimulated β-cells following treatment with 10 µM psilocybin. INS-1 β-cells underwent a 2 h pre-treatment with psilocybin, followed by a 24 h incubation in HG-HL conditions (250 μM palmitic acid + 25 mM glucose). Total RNA was extracted from both treated and untreated cells and utilized for qRT-PCR. The figure illustrates the normalized mRNA expression levels of *Ins1*, *Ins2*, *PDX-1*, *FOXO1*, *NEUROD1*, *MafA*, *Slc2A2*, *Nanog*, and *Pou5f1* in both HG-HL-stimulated and unstimulated β-cells treated with 10 µM psilocybin. The results are shown as the average value with the standard deviation, from three replicates (N = 3). Abbreviations include Ct (control), PSI (psilocybin), and HG-HL (high glucose + high lipid). Significance is marked by asterisks: one for *p* < 0.05, two for *p* < 0.01, three for *p* < 0.001, four for *p* < 0.0001, and “ns” for non-significant outcomes.

## Data Availability

The original Western blot images were provided to the *Genes* journal.

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
