# Peer review of "The Impact of Psilocybin on High Glucose/Lipid-Induced Changes in INS-1 Cell Viability and Dedifferentiation"

_genes, 2024, doi:10.3390/genes15020183_

Round 1
Reviewer 1 Report
Comments and Suggestions for Authors
The authors provided a well written comprehensive report detailing findings examining a naturally derived Psilocybin on cell survival, dedifferentiation, and function using the INS-1 832/13 rat Insulinoma cell line. Findings are novel and certainly provide relevant initial in-vitro data on the potential therapeutic use of Psilocybin on type 2 diabetes.
Major comments:
1. The authors should provide a detailed statement on their source or supplier of Psilocybin. I could not find this description in the materials and methods.
2. The authors should provide a description of the INS-1 832/13 cell line.
3. Along the lines of 2 above, the the INS-1 832/13 cell line contains a human insulin expression cassette that allows for human insulin secretion. This cell line also produces rat insulin. The ELISA utilized only detected rodent but not human insulin. Asking if the authors considered evaluating the human insulin secretion as well since they showed that Psilocybin did not exhibit any positive effects on the impaired GSIS in HG-HL-stimulated β-cells.
4. Did the authors examine intracellular insulin content? and consider expressing data as secretion relative to insulin content.
5. Experiments employing use of murine or human islets would greatly enhance significance of findings but certainly not necessary in this report. However this can be mentioned in limitations and future directions.
Minor comment:
1. The title has an atypical grammatical structure and does not precisely state what was measured or examined. Cell loss was not determined but rather viability. Also, primary beta cells were not examined but rather cell line.
Would recommend title change to:
The impact of Psilocybin on High Glucose/Lipid-induced INS-1 Cell Viability and Dedifferentiation
2. Subheading of 3.1. 10 µM Psilocybin Mitigates HG-HL-Induced β-Cell Loss should have loss removed and replaced with "reduced viability"
3. Subheading of 3.2 has apoptotic misspelled
Comments on the Quality of English Language
Overall, manuscript was well written and described.
Author Response
Major comments
- The authors should provide a detailed statement on their source or supplier of Psilocybin. I could not find this description in the materials and methods.
Answer: The psilocybin used in this study (CAS No. 520-52-50) was provided by Applied Pharmaceutical Innovation, Edmonton, AB, Canada, and its source is appropriately incorporated in the paper.
- The authors should provide a description of the INS-1 832/13 cell line.
Answer: The following information was incorporated in the paper “The INS-1 832/13 Rat Insulinoma Cell line (Catalog Number: SCC207, sourced from EMD Millipore Corporation, Temecula, CA, USA) is a variant of INS-1, selected for its strong Glucose-Stimulated Insulin Secretion (GSIS) capabilities. It features a human insulin expression cassette, facilitating the production and release of both rat and human insulin. The presence of a human insulin expression cassette ensures the consistent secretion of human insulin over extended passages under selective conditions. The INS-1 832/13 cellular model also serves as a valuable system for exploring the intricacies of cellular insulin release, storage, and production mechanisms.”
- Along the lines of 2 above, the INS-1 832/13 cell line contains a human insulin expression cassette that allows for human insulin secretion. This cell line also produces rat insulin. The ELISA utilized only detected rodent but not human insulin. Asking if the authors considered evaluating the human insulin secretion as well since they showed that Psilocybin did not exhibit any positive effects on the impaired GSIS in HG-HL-stimulated β-cells.
Answer: It would be a valuable suggestion to assess human insulin secretion alongside rat insulin. It is noteworthy that the Rat/Mouse Insulin ELISA kit (EZRMI-13K) exhibits 100% cross-reactivity with both rat and human insulin. The cross-reactivity information for the kit, sourced from Millipore, can be found in the following image and on their website (https://www.emdmillipore.com/CA/en/product/Rat-Mouse-Insulin-ELISA,MM_NF-EZRMI-13K#documentation).
- Did the authors examine intracellular insulin content? and consider expressing data as secretion relative to insulin content.
Answer: We attempted to quantify intracellular insulin content using an ELISA kit. However, due to the substantial amount of internal insulin, even after a 500-fold dilution, we were unable to measure the intracellular insulin content.
- Experiments employing use of murine or human islets would greatly enhance significance of findings but certainly not necessary in this report. However, this can be mentioned in limitations and future directions.
Answer: The suggestion has already been incorporated into the 'Limitations and Future Directions' section of the paper.
Minor comments
- The title has an atypical grammatical structure and does not precisely state what was measured or examined. Cell loss was not determined but rather viability. Also, primary beta cells were not examined but rather cell line.
Would recommend title change to:
The impact of Psilocybin on High Glucose/Lipid-induced Changes in INS-1 Cell Viability and Dedifferentiation
Answer: The title has been modified accordingly
- Subheading of 3.1. 10 µM Psilocybin Mitigates HG-HL-Induced β-Cell Loss should have loss removed and replaced with "reduced viability."
Answer: The modification has been incorporated accordingly.
- Subheading of 3.2 has apoptotic misspelled
Answer: The modification has been implemented accordingly.

Reviewer 2 Report
Comments and Suggestions for Authors
The manuscript investigated psilocybin administration effectively mitigates HG-HL-induced β-cell loss, potentially mediated through the modulation of apoptotic biomarkers, which is possibly related to the mitigation of TXNIP and STAT-1 and STAT-3 phosphorylation. This topic is interesting and the manuscript is well organized; however, there are some mirror questions:
1. Why choose 10 μM for the dose of Psilocybin? Please give a reasonable explanation.
2. There were only RT-PCR results for the factors Slc2A2, NeuroD1, MafA, Nanog and Pou5f1, but no WB results. Please verify that psilocybin did not show any significant effects on the factors by WB test.
3. The effects of psilocybin on cell cycle should investigated.
Author Response
- Why choose 10 μM for the dose of Psilocybin? Please give a reasonable explanation.
Answer: The selection of this dose was done based on the results of pilot studies conducted with INS-1 and other cell lines before the main trials. Considering the potential adverse side effects of psilocybin, our aim was to identify the lowest dose that maintains the highest effectiveness. An explanation regarding the rationale for choosing this specific dose has been integrated into the paper.
- There were only RT-PCR results for the factors Slc2A2, NeuroD1, MafA, Nanog and Pou5f1, but no WB results. Please verify that psilocybin did not show any significant effects on the factors by WB test.
Answer: The exploration of Slc2A2, NeuroD1, MafA, Nanog, and Pou5f1 responses to psilocybin at the protein levels represents a limitation of this study. This limitation has been incorporated in the 'Limitations and Future Directions' section.
- The effects of psilocybin on cell cycle should investigated.
Answer: Considering the potential influence of serotonin receptor agonists on beta cell proliferation, studying the impact of psilocybin on the cell cycle is a valuable suggestion. It's important to note that we have indeed conducted a cell cycle assay; however, the results have not been included in the paper. The preliminary findings of the cell cycle assay are outlined below:
The HG-HL conditions, used to induce beta cell dedifferentiation (250 µM PA + 25 mM Glucose for 24 hours), have been observed to increase the S phase of the cell cycle in beta cells as compared to control groups. This response may be attributed to the initial reaction of beta cells to hyperglycemia, leading to the propagation of beta cells to compensate for the increased need for insulin during the insulin resistance phase. The figure indicates that psilocybin further increases the proliferation of beta cells compared to the HG-HL conditions. To validate these preliminary results, it is essential to investigate the response of various biomarkers involved in different cell cycle phases. We plan to explore this in the future.

Reviewer 3 Report
Comments and Suggestions for Authors
The study investigates the impact of psilocybin on the survival, dedifferentiation, and function of insulin-secreting cells (β-cells) under high glucose-high lipid conditions, revealing its effective mitigation of cell loss and modulation of relevant genes, providing a foundation for its potential application in Type 2 diabetes treatment.
1.This study solely conducted cell experiments and lacked animal experiments.
2.The research focused only on protein and gene expression analysis, lacking evidence from relevant fluorescent electron microscopy imaging experiments.
3.There is a scarcity of references to recent literature within the past 3 years.
4.In the discussion section, please elucidate the mechanism of action of psilocybin on high glucose-high lipid-induced β-cell loss and dedifferentiation.
5.Further modifications are required in terms of article format and language.
Comments on the Quality of English LanguageFurther language polishing is required in order to meet the publication requirement .
Author Response
1.This study solely conducted cell experiments and lacked animal experiments.
Answer: As emphasized in the "Limitations and Further Directions" section of the paper, the absence of animal experiments can be regarded as the primary limitation of this study. This in vitro investigation, marking the first exploration into the impact of psilocybin on diabetes-related factors, may pave the way for further investigations into the potential anti-diabetic effects of psilocybin.
2.The research focused only on protein and gene expression analysis, lacking evidence from relevant fluorescent electron microscopy imaging experiments.
Answer: The suggestion of using fluorescent electron microscopy imaging experiments is valuable for studying cell and organelle morphology. However, our research focuses on exploring the effects of psilocybin on HG-HL-induced Ins-1 cell viability reduction, dedifferentiation, and dysfunctionality. To achieve this, we utilized the MTT assay, Western blotting (WB), Caspase3/7 activity assay, quantitative real-time polymerase chain reaction (qRT-PCR), and enzyme-linked immunosorbent assay (ELISA).
While this technique could potentially be employed to study the intracellular calcium content, a crucial factor in insulin secretion in beta cells, we did not observe a significant impact of psilocybin on impaired Glucose-Stimulated Insulin Secretion (GSIS) in HG-HL-induced beta cells. Therefore, investigating the influence of psilocybin on the intracellular calcium content in the HG-HL-induced INS-1 cell line did not seem to be warranted.
3.There is a scarcity of references to recent literature within the past 3 years.
Answer: In this regard, it should be mentioned that investigation of pharmaceutical properties of psilocybin is a new area, leading to the general scarcity of relevant literatures. As far as we know, the current research is the first study regarding the potential impact of psilocybin on the mitigation of T2DM.
Some references have been updated with more recent ones.
4.In the discussion section, please elucidate the mechanism of action of psilocybin on high glucose-high lipid-induced β-cell loss and dedifferentiation.
Answer: To unravel the mechanism of psilocybin's action on HG-HL-induced beta cell loss and dedifferentiation, it is imperative to investigate the response of numerous other proteins involved in the signal transduction between serotonin receptors and the target proteins associated with beta cell loss and dedifferentiation to psilocybin. In our current study, we identified STAT1, STAT3, and TXNIP as potential downstream targets for psilocybin, contributing to the mitigation of apoptotic responses. The IL-6/JAKs axis typically facilitates the phosphorylation and subsequent activation of STATs. To determine whether this axis is implicated in the mitigative effects of psilocybin on apoptosis, it is recommended to initially study the response of this axis to psilocybin in HG-HL-induced beta cells. As mentioned in the limitations section of the paper, in general, exploring the response of serotonin receptors and downstream signaling cascades is advised. A suggestion regarding this has been added to “future directions” section of the paper.
5.Further modifications are required in terms of article format and language.
Answer: We checked the language and the format.

Reviewer 4 Report
Comments and Suggestions for Authors
Major Revision
1. Is there any specific reason to use 10 uM of psilocybin? Any literature review, and or other pilot study.
2. As it is described that psilocybin interact with serotonin receptors in the brain, do psilocybin also interact with serotonin receptor in beta cells? It is preferable if you provide some literature reference or some data that psilocybin interacts with serotonin receptor in INS-1 cells.
3. If administration of psilocybin alleviates HG-HL-induced β-cell loss, through apoptotic biomarkers, TXNIP and STAT-1 and STAT-3 phosphorylation, can inhibition of serotonin receptor or inhibition of apoptosis/ STAT-1 and STAT-3 reverse the effect of psilocybin?
Minor Revision
4. In Figure 6, mRNA expression of FoxO1 between HGHL and HGHL+PSI 10 is a big difference, but it shows no significance. Confirmation should be made.
5. In Figure 6, the digits on the graph of Nanog and Pou5f1 seems awkward. You need to double check again.
6. The quality of the figure needs to be improved and some rearrangement should be made to improve the quality of the article.
7. Indications for figure should be addressed correctly. For example, In Line 273, Line 287, line 294, Line 299. Line 318, it was wrongly addressed.
Author Response
Major Revision
- Is there any specific reason to use 10 uM of psilocybin? Any literature review, and or other pilot study.
Answer: The selection of this dose was done based on the results of pilot studies conducted with INS-1 and other cell lines before the main trials. Considering the potential adverse side effects of psilocybin, our aim was to identify the lowest dose that maintains the highest effectiveness. An explanation regarding the rationale for choosing this specific dose has been integrated into the paper.
- As it is described that psilocybin interact with serotonin receptors in the brain, do psilocybin also interact with serotonin receptor in beta cells? It is preferable if you provide some literature reference or some data that psilocybin interacts with serotonin receptor in INS-1 cells.
Answer: There are multiple reports discussing the effects of serotonin, which serves as a natural agonist of serotonin receptors, on the proliferation and activity of beta cells. Notable references include numbers 9, 10, 11, and 12 in the papers. Given the well-established interaction of psilocybin with serotonin receptors, particularly SR-2A and SR-2B, and the presence of serotonin receptors on beta cells, it can be assumed that psilocybin may bind to these receptors on beta cells. However, as far as we know, there is no report confirming this so far. It should be mentioned that this paper represents the initial report on the potential role of psilocybin in mitigating diabetes, numerous aspects of the molecular impacts of psilocybin on beta cells remain unknown.
- If administration of psilocybin alleviates HG-HL-induced β-cell loss, through apoptotic biomarkers, TXNIP and STAT-1 and STAT-3 phosphorylation, can inhibition of serotonin receptor or inhibition of apoptosis/ STAT-1 and STAT-3 reverse the effect of psilocybin?
Answer: In the present study, we observed that the administration of psilocybin mitigates HG-HL-induced β-cell loss. If serotonin receptors serve as direct targets for psilocybin in this process, inhibiting serotonin receptors may counteract the effects of psilocybin. This hypothesis suggests that the beneficial impact of psilocybin could be mediated through serotonin receptors. Additionally, it should be noted that, apart from binding to serotonin receptors, psilocybin could modulate the glutamate system, which is also present in beta cells (DOI: 10.1038/s41386-020-0718-8), inhibit the serotonin transporter site (DOI: 10.1038/s41386-023-01648-7), and affect the levels of serotonin receptors (as indicated in figure 1, unpublished data).
Figure 1. The western blot analysis of SR-2B in response to 10 mM psilocybin in HG-HL-induced beta cells. Psilocybin treatment reduces the level of SR-2B protein in HG-HL-induced beta cells.
If apoptosis is the sole mechanism through which psilocybin confers its protective effects on beta cell viability, inhibiting apoptosis might mimic the impact of psilocybin on beta cell viability. However, it is important to consider other mechanisms such as pyroptosis that may be involved in improving beta cell viability. To comprehensively address this question, the effects of psilocybin on all mechanisms influencing the viability and proliferation of beta cells should be investigated. In addition, if TXNIP, STAT1, and STAT3 are not the exclusive targets for psilocybin in exerting its mitigative effects on apoptosis, their inhibition could potentially mimic the impact of psilocybin on beta cell viability. To address this question comprehensively, an in-depth study of the response of all genes/proteins involved in apoptosis to psilocybin in HG-HL-induced beta cells is essential.
Minor Revision
- In Figure 6, mRNA expression of FoxO1 between HGHL and HGHL+PSI 10 is a big difference, but it shows no significance. Confirmation should be made.
Answer: The figure has been reviewed and confirmed. It is worth noting that the choice of analysis method can influence the results. In the presented figure (figure 2), the Duncan test was employed to compare the HG-HL group with others, revealing a significant difference between HG-HL and HG-HL+Psi 10. However, when we utilized Tukey’s test for comparing all groups, as outlined in the paper, the results varied. This could be attributed to the elevated standard deviation observed in the Ct and Ct+Psi10 groups
Figure 2. The qRT-PCR analysis of FOXO1 in beta cells in response to psilocybin under HG-HL-induced and uninduced conditions. The Duncan test was used to compare the HG-HL group with other groups.
- In Figure 6, the digits on the graph of Nanog and Pou5f1 seems awkward. You need to double check again.
Answer: The digits were double-checked and confirmed. The raw normalized data is as follows:
|
Nanog |
|
|
|
|
Ct |
2.06456977 |
0.52016963 |
0.93173166 |
|
Ct+PSi10 |
6.87802648 |
4.6586387 |
7.39558627 |
|
HG-HL |
80.6685223 |
103.493692 |
64.3715504 |
|
HG-HL+PSi10 |
8.29940156 |
5.72510039 |
3.64046739 |
|
Pou5f1 |
|
|
|
|
Ct |
2.13502793 |
2.18313344 |
3.30881179 |
|
Ct+PSi10 |
6.93753657 |
6.95783341 |
6.53462874 |
|
HG-HL |
91.8848529 |
80.1334052 |
103.199149 |
|
HG-HL+PSi10 |
4.13720826 |
5.88838471 |
5.67712265 |
- The quality of the figure needs to be improved and some rearrangement should be made to improve the quality of the article.
Answer: If there are specific figures that require improvement or modification, kindly identify them so that we can proceed with the necessary enhancements.
- Indications for figure should be addressed correctly. For example, In Line 273, Line 287, line 294, Line 299. Line 318, it was wrongly addressed.
Answer: The modifications have been introduced accordingly.

Round 2
Reviewer 3 Report
Comments and Suggestions for Authors
The research content is not sufficiently comprehensive, and we hope to further improve it in future research.
Comments on the Quality of English LanguageNo obvious language issues were found.
Author Response
Thanks; we hope the manuscript is more clear now.

Reviewer 4 Report
Comments and Suggestions for Authors
1. If concentration of psilocybin was chosen based on the results of pilot studies, data should provide in the article for the convenience of readers.
2. The data of SR-2B in response to psilocybin in HG-HL-induced beta cells should provide in article or as supplementary data.
3.The effect of psilocybin should be confirmed by inhibiting serotonin receptor or inhibiting STAT3.
Author Response
- If concentration of psilocybin was chosen based on the results of pilot studies, data should provide in the article for the convenience of readers.
Answer: As outlined in the paper, our selection was informed by our laboratory findings with various In vitro systems, including THP-1 macrophages (unpublished data), and human 3D EpiIntestinal tissue (recently published, https://doi.org/10.3390/life13122345). We now referred to this published data for the decision on dose selection.
Notably, some of the unpublished data regarding THP-1 macrophages is presented below:
Figure 1) The Western blot analysis illustrates the response of IL-1Beta, IL-6, and P-NFkB in LPS-induced THP-1 macrophages to three psilocybin doses: 5, 10, and 15 μM.
As previously noted, our goal was to identify the minimum effective dose of psilocybin with optimal pharmaceutical properties. It is noteworthy that the dysregulation of IL-6, IL-1beta, and P-NF-kB significantly contributes to the onset of diabetes. We could not include these data since it was obtained from experiments on different cell type.
- The data of SR-2B in response to psilocybin in HG-HL-induced beta cells should provide in article or as supplementary data.
Answer: The findings related to SR-2B have been included in the supplementary data, accompanied by an explanatory section added to the paper.
- The effect of psilocybin should be confirmed by inhibiting serotonin receptor or inhibiting STAT3.
Answer: Thank you for the valuable suggestion. A corresponding suggestion has been incorporated into the forthcoming section of the paper. Since this work is not focused on the description of serotonin receptor-dependence of the effect of psilocybin on HG-HL challenged beta cells, these experiments are out of scope of the paper. We will definitely plan to perform these experiments in our future work.

Round 3
Reviewer 4 Report
Comments and Suggestions for Authors
In Figure 1, 4 and 6, for non-significant data, describing "ns" in the graph is not necessary.
The paper is well written.